



# Gradient Boosting Machine Learning to Improve Satellite-Derived Column Water Vapor Measurement Error

Allan C. Just[1][*], Yang Liu[1], Meytar Sorek-Hamer[2,3], Johnathan Rush[1], Michael Dorman[4], Robert Chatfield[3], Yujie Wang[5,6], Alexei Lyapustin[6], Itai Kloog[4]

[1]Icahn School of Medicine at Mount Sinai, New York, USA
[2]Universities Space Research Association (USRA), California, USA
[3]NASA Ames Research Center, California, USA
[4]Ben-Gurion University of the Negev, Beer Sheva, Israel
[5]University of Maryland Baltimore County, Maryland, USA
[6]NASA Goddard Space Flight Center, Maryland, USA

*Correspondence to: Allan C. Just (*allan.just@mssm.edu)

**Abstract.** The atmospheric products of the Multi-Angle Implementation of Atmospheric Correction (MAIAC) algorithm include column water vapor (CWV) at 1 km resolution, derived from daily overpasses of NASA's Moderate Resolution Imaging Spectroradiometer (MODIS) instruments aboard the Aqua and Terra satellites. We have recently shown that machine learning using extreme gradient boosting (XGBoost) can improve the estimation of MAIAC aerosol optical depth (AOD). Although MAIAC CWV is generally well validated (Pearson's R >0.97 versus CWV from AERONET sun photometers), it has not yet been assessed whether machine-learning approaches can further improve CWV. Using a novel spatiotemporal cross-validation approach to avoid overfitting, our XGBoost model with nine features derived from land use terms, date, and ancillary variables from the MAIAC retrieval, quantifies and can correct a substantial portion of measurement error relative to collocated measures at AERONET sites (26.9% and 16.5% decrease in Root Mean Square Error (RMSE) for Terra and Aqua datasets, respectively) in the Northeastern USA, 2000-2015. We use machine-learning interpretation tools to illustrate complex patterns of measurement error and describe a positive bias in MAIAC Terra CWV worsening in recent summertime conditions. We validate our predictive model on MAIAC CWV estimates at independent stations from the SuomiNet GPS network where our corrections decrease the RMSE by 19.7% and 9.5% for Terra and Aqua MAIAC CWV. Empirically correcting for measurement error with machine-learning algorithms is a post-processing opportunity to improve satellite-derived CWV data for Earth science and remote sensing applications.

## 1 Introduction

Water vapor represents a small but environmentally significant constituent of the atmosphere. The integrated water vapor from ground to space is defined as the column water vapor (CWV), in units of centimeters (i.e. precipitable water vapor) (Gao and Goetz, 1990). CWV has important applications in many fields, such as atmospheric correction of remote sensing images, Earth energy balance and global climate change, land surface temperature retrieval in thermal remote sensing, and astronomy. Thus, high resolution CWV values with global coverage have multiple uses in Earth science and remote sensing. CWV has been measured by multiple technologies and



monitoring networks, including sun photometers, GPS sensors (e.g. SuomiNet), Aerosol Robotic Network (AERONET) sun photometers, and satellite remote-sensing. The AERONET sun photometer network measures CWV in approximately 400 stations worldwide, in channels centered at 940nm and provided to the user in Level 2, which is the highest data-quality level provided by AERONET (Pérez-Ramírez et al., 2014). The AERONET CWV data have been well validated with the U.S. Department of Energy Atmospheric Radiation Measurement Program

(ARM) radiosonde observations and other ground-based retrieval techniques such as microwave radiometry (MWR) and SuomiNet GPS receivers, and do not observe any dependence of biases with the zenith angle (Pérez-Ramírez et al., 2014). AERONET CWV has been used in studies that examine aerosol optical, microphysical, and radiative properties in Africa (Adesina et al., 2014; Boiyo et al., 2019; Kumar et al., 2013), in the Brazilian tropics (Schafer et al., 2008) , and in Beijing and Kanpur (Wang et al., 2011). Global satellite-borne CWV is available at high

resolution (1 km), from the Multi-Angle Implementation of Atmospheric Correction (MAIAC) algorithm derived from daily overpasses of NASA's Moderate Resolution Imaging Spectroradiometer (MODIS) instruments aboard the Aqua and Terra satellites. The MAIAC CWV is computed using MODIS near-infrared (NIR) channels centered at 940nm. This method applies two ratios of channels to compute the water vapor transmittance, and then compute the amount of water vapor using look-up-tables (Lyapustin et al., 2014). The MAIAC CWV algorithm was validated

against ground measurements of CWV from 265 AERONET stations worldwide, with a relatively strong association (Pearson's R > 0.95; root mean squared error [RMSE] < 0.25 cm; average accuracy of ±15%) (Martins et al., 2019). These datasets were collocated by averaging MAIAC values within 9 x 9 pixels and AERONET values ±30 minutes of the satellite overpass in cloud-free conditions. A significant upward trend (p < .05) for MAIAC TERRA was found over most regions, although this was not significant over the Northeastern USA. Globally, the highest average

correlation between MAIAC CWV retrievals from both Aqua and Terra with AERONET CWV have been shown in Asia and both Northern and Southern regions of the USA.

     In spite of the strong performance of MAIAC CWV in multiple locations, comparing it with collocated AERONET CWV, there may be opportunities to characterize and correct complex interactions and challenging conditions that

increase satellite retrieval error. However, it has not yet been assessed whether machine-learning approaches can improve the estimation of satellite-borne CWV. We have recently demonstrated that machine learning using eXtreme Gradient Boosting (XGBoost) can improve the estimation of MAIAC aerosol optical depth (AOD) parameters over AERONET stations (43% decrease in cross-validated RMSE) (Just et al., 2018). Gradient boosting involves fitting a large number of tree-based models where each subsequent tree is made a weak predictor of the

error from the previous trees. The component trees use a recursive binary partitioning that accommodates varying types and scales of predictor variables and is robust to outliers (Elith et al., 2008). An advantage of flexible algorithmic machine learning approaches such as XGBoost is that they can model complex phenomena, including interactions of multiple features (e.g. retrieval angles, seasonality, and surface characteristics). The resulting prediction model can provide an algorithm to reduce the retrieval error. Machine-learning tools for model

interpretation can also help explain the contributions of these features to retrieval error and guide feature selection to build parsimonious models.



While the satellite data record continues to grow, the ground monitoring networks that can be used for validation and algorithmic measurement error correction of satellite retrieval products are still sparse. Collocated ground-satellite datasets may thus have important non-independent spatiotemporal structure if they rely on observations that occur in only a few locations. Flexible machine-learning models would overfit to the characteristics of these particular stations or the days when AERONET data are available if cross-validation assumed independence of observations. While machine-learning applications in aerosol research have begun to adopt group-K-fold cross validation for assessing model fit across fixed monitoring networks (Di et al., 2016), we propose a novel cross-validation approach taking into consideration data structure due to both fixed sites and correlation of observations from the same day.

The goals of this work are to 1) evaluate whether machine-learning gradient boosting models can improve satellite-based CWV retrievals, and 2) understand the contributions of different features as well as spatial and temporal structures of the ground station measures to predictions of error in the estimated CWV. The data and machine-learning methods are described in section 2, followed by a discussion of the results in sections 3 and 4.

## 2 Description of the Data

In order to assess the agreement of the MAIAC estimates of CWV with those from AERONET, collocated datasets were built using MAIAC data where AOD was available (representing clear sky conditions) from both Terra and Aqua (separately) collocated to the nearest 1 km * 1 km grid centroid and the closest observation in time (no more than 60 minutes) with cloud-screened (version 2, level 2.0) (Smirnov et al., 2000) measures of CWV from the AERONET network of sun photometers over the Northeastern USA (including 13 states and the District of Columbia from Maine to Virginia). The study period included 10,247 observations (from 75 AERONET stations) for Terra (2000-2015) and 8,536 observations (from 71 stations) for Aqua (2002-2015). All analyses were performed for Terra and Aqua datasets separately. AERONET stations in the Northeast are largely urban and coastal (Fig. 1). We defined our target modeling parameter as the difference between MAIAC and AERONET CWV (ΔCWV = MAIAC CWV - AERONET CWV) such that any variation from zero indicated a component of measurement error that we sought to explain.

After exploratory scatterplots of ΔCWV versus time showed a temporal cluster of large outliers coming from a single AERONET station (City College of New York), observations from this site between 2007-06-17 and 2009-01-01 were dropped from further analysis, including 99 observations collocated for Terra and 95 observations for Aqua datasets. This particular period, which was flanked on both sides with months without values at that station, showed a clear deviation from the monitor's typical trend across the remainder of the study period.





The date range for the collocated Terra dataset was from 2000-02-25 to 2015-12-27, including observations from 3,024 unique days (52% of days during this interval). The date range for the collocated Aqua dataset was from 2002-07-04 to 2015-12-28, including observations from 2,627 unique days (53% of days during this interval).

## 3 Statistical Methods

We examined the use of XGBoost (Chen and Guestrin, 2016), for improving satellite-based MAIAC CWV retrievals, and decreasing estimation error as this method had previously outperformed two related supervised learning approaches using regression trees, namely gradient boosting and random forests, in a similar application (Just et al., 2018). The XGBoost algorithm is a popular implementation of boosted regression trees (Friedman, 2001). A regression tree is a model that specifies a few recursive binary splits of predictors and assigns a constant

value to all cases that end up in the same terminal node. A set of multiple trees can be used for prediction by combining the outputs of the individual trees for each case. Such a set of trees can accommodate complex relationships including non-linearities and interactions while being robust to outliers. Boosting is a method of fitting a series of models iteratively, with each model fit on the residuals of the previous models. While each tree is individually a weak learner, this process of sequentially learning by combining many iteratively fit trees performs

well, achieving low testing error. The XGBoost package is a scalable gradient boosting implementation with additional features including penalties to avoid overfitting and optimized computational speed (Chen and Guestrin, 2016).

We end up with more parsimonious XGBoost models, i.e. fewer trees, by adopting the concept of 'dropout' from

deep learning, in which individual learners are randomly dropped during training. XGBoost models were fit with tree dropout using DART: Dropout meets Additive Regression Trees (Vinayak and Gilad-Bachrach, 2015). The tree dropout implementation in DART avoids the diminishing contributions and over-specialization of later trees in gradient boosted regression trees. This is particularly important in our application given the low number of AERONET stations and relatively small size of the collocated datasets for machine-learning algorithms. XGBoost

has several hyperparameters related to the desired size and complexity of the model that need to be set in training for each dataset. Each time we trained a model we further split the training data and performed a nested comparison of XGBoost models using 50 random sets of potential hyperparameters generated to be well-spaced across the range of potential hyperparameter values using Latin hypercube sampling. We selected the set of hyperparameters that minimized the RMSE within the withheld portion of the training data before refitting with all training data.


Prior to feature selection, initial analyses included 25 candidates features such as: MAIAC variables including an uncertainty parameter related to blue band surface reflectance, relative azimuth angle, and AOD; time trend (integer date); elevation; several land use terms from the National Land Cover Database 2011 aggregated to proportions within 1 km * 1 km grid cells as well as proportion of water within 5-15 km buffers; and distance to major water

bodies (Great Lakes and the Atlantic Ocean). Feature engineering calculated candidate features based on spatial



patterns in non-missing MAIAC data including the number of contiguous non-missing grid cells (clump size) and the number of non-missing observations in focal windows of side lengths from 30-510 km. Details on the data sources and feature engineering for all candidate features are included in Supplementary Materials. No external meteorology or assimilated data were included.


The contributions of each feature to cross-validated predictions were estimated from Shapely Additive Explanations (SHAP) values (Lundberg et al., 2018). These SHAP values form an additive feature attribution measure to interpret complex machine-learning models. SHAP values estimate the contributions of each feature to each individual prediction (for $\Delta$CWV, this is in units of cm). Specifically, the SHAP value for a given predictor and a given

observation is the difference in the output, i.e. a predicted $\Delta$CWV, if the model is fit with or without the predictor. For each observation, the sum of all SHAP values, plus the bias term (the overall mean of $\Delta$CWV), equals the prediction from the XGBoost model. The resulting matrix of SHAP values can be summarized to understand how a predictor contributes to the predictions. The mean absolute SHAP value across all observations summarizes the global feature importance and more local model interpretation is possible through exploratory data visualizations

such as scatterplots of individual predictors versus their SHAP values.

Because a more parsimonious set of features can ease future efforts to build large spatiotemporal datasets for algorithmic correction, an initial feature selection approach was performed prior to evaluating overall model performance. Feature selection was performed in a randomly selected 20% subset of the data to avoid overfitting

prior to later model evaluation steps. Within this subset, we evaluated both the mean absolute SHAP values as a measure of global feature importance within a full model with all 25 candidate features, as well as a recursive stepwise procedure. We adopted 5-fold cross-validation split by MAIAC stations to alleviate overfitting to spatial features of the relatively low number of unique stations. In each round of cross-validation, backwards feature selection was applied to rank and remove the features by increasing importance. Starting with the XGBoost model

containing all 25 candidate features, the overall RMSE was calculated from the out-of-sample predictions after cross-validation. Then the feature importance was ranked by mean absolute SHAP values for all the features in the model from low to high. This step was repeated removing the least important feature at each step. After plotting the overall RMSE from the cross-validated predictions against the number of features, we selected the model with the lowest RMSE for Aqua and Terra separately. We then pooled the set of top-ranked features from both satellites to

facilitate comparisons between the Aqua and Terra models examined in the full dataset.

Using the selected features, grouped ten-by-ten-fold cross-validation randomly splitting the data by both station and day was performed on the whole dataset. In each training iteration, all observations from one fold of stations and from one fold of days were withheld with the remaining dataset containing roughly 0.9*0.9 = 81% of the training

data (a similar share of training data as in 5-fold cross-validation). However, for each combination of withheld data, predictions for evaluating model performance and the corresponding SHAP values were only made in the intersection of withheld days and monitors (~1% of the data). Thus predictions for each observation were made on a





model trained without any observations from the same day or station (see Fig. 2). For comparison, we also evaluated

model performance using grouped-5-fold cross-validation separately splitting the data by station or by day.

Hyperparameter tuning of the XGBoost model was performed separately in each round of cross-validation.

While we used an aggregated measure of the mean absolute SHAP value for each feature as a measure of feature importance in our variable selection, we also plotted the out-of-sample SHAP in order to aid model interpretability. In particular, we plotted frequencies of SHAP values by variable and in bivariate scatterplots versus observed

values.

Finally, we conducted an additional external validation of our final model by comparing both the original MAIAC CWV and our corrected CWV with an independent dataset of CWV measured by GPS-based stations in the SuomiNet dataset (Ware et al., 2000), within our Northeastern USA study region - many of which are quite distant

from the AERONET sites. All SuomiNet stations use precision survey-quality dual-frequency GPS receivers and antennas. The water-lag derived CWV measures from GPS-based stations are generally considered to have excellent precision (5-10%), exceeding that from sun photometers (Pérez-Ramírez et al., 2014).

## 4 Results

### 4.1 Descriptive analysis of CWV and ΔCWV

The overall agreement of the original MAIAC CWV and AERONET CWV was quite good with a Pearson's correlation of 0.976 and 0.984 for Terra and Aqua, respectively, in agreement with the global MAIAC CWV validation (Martins et al., 2019). However, outlying values and a positive bias in Terra-derived MAIAC CWV particularly indicate a potential for improvement in MAIAC CWV relative to AERONET. The target parameter of ΔCWV (based on the difference between MAIAC and AERONET) was approximately symmetrically distributed

and had a mean of 0.043 cm and -0.054 cm, and a standard deviation of 0.25 cm and 0.18 cm for the collocated Terra and Aqua datasets, respectively (Table 1). Descriptive scatterplots of the ΔCWV versus individual predictors showed some clear patterns prior to modeling (Fig. 3 and Supplementary Fig. S1). For example, there is a clear seasonal pattern with a larger SD of ΔCWV in the summer (0.32 cm and 0.25 cm for Terra and Aqua) when the SD of AERONET CWV is also highest (0.90 cm). This seasonal pattern and the positive bias for Terra (MAIAC CWV

overestimates AERONET CWV) is seen to grow larger in more recent years (e.g. 2010-2015). This trend is related to the trend in MODIS Terra calibration, as previously reported (Martins et al., 2017).

### 4.2 Feature Selection and Model Performance

Variable selection using feature importance from SHAP was run in a 20% subset for both Terra and Aqua datasets.

Using both global feature importance from a full model and a stepwise backward selection calculating RMSE at





each step after ranking variable importance by mean absolute SHAP, we selected 6 features for the Terra model and selected 7 features for the Aqua model. The 4 features shared by both models were time trend (date represented as an integer), MAIAC CWV, blue band uncertainty, and MAIAC AOD. The other variables selected for Terra were elevation and distance to major water body, and for Aqua were the proportion of forest in a 1x1 km square, relative

azimuth angle, and the proportion of developed area in a 1x1 km square. Pooling these features from both satellites brought the original set of 25 features down to a more parsimonious set of 9 with little loss of model performance (results not shown).

Using the reduced feature set, we implemented the cross-validation in the full dataset to evaluate model

performance. In the collocated Terra dataset, the predicted ΔCWV evaluated with the grouped monitor-by-day cross-validation (10*10 fold) explained 45.0% ($R^2$) of the variance in ΔCWV and reduced the RMSE from 0.252 cm (the root mean squared difference between MAIAC and AERONET CWV) to 0.184 cm, a 26.9% decrease in RMSE. In the collocated Aqua dataset, the predicted ΔCWV explained 24.1% of the variance in ΔCWV ($R^2$) and reduced the RMSE from 0.189 cm to 0.158 cm, a 16.5% decrease in RMSE.


The evaluation of model performance was substantively different depending on how the cross-validation strategy reflected the data structure. Ignoring the non-independence of the training data by site and withholding unique days for grouped 5-fold cross-validation (training on 80% of the data), RMSE for Terra was 0.146 and for Aqua was 0.145 (Table 2), a much better performance (smaller RMSE) that indicates over-fitting to the particular sites in the

training dataset. Similarly, the RMSE from cross-validation split by station (and not by day) was also slightly lower than the RMSE from station-by-day cross-validation suggesting a much smaller degree of overfitting also to the specific dates in the training set.

After applying the XGBoost model, the measurement error of ΔCWV was corrected to be closer to zero, particularly

for the largest magnitude ΔCWV values. For Terra and Aqua respectively, 87% and 93% of the ΔCWV observations beyond one standard deviation (outside of the dotted lines in Fig. 4, making up 24% of the collocated observations in Terra and 19% in Aqua) had lower measurement error (|ΔCWV|) by an average magnitude of 41% smaller in Terra and 53% smaller in Aqua after XGBoost correction.

We describe the variation in hyperparameters from XGBoost models across the 100 runs of the site-by-day 10*10 fold cross-validation. Greater variation in the selected hyperparameter values across folds with very similar training datasets may indicate a lower impact on model performance (Supplementary Table S2).

### 4.3 Variable Importance Assessment

Although the final model had already been restricted to include only the top variables from our variable selection approach, we further interpreted variable importance and the contribution of these variables with SHAP values





estimated in the grouped-cross-validation (at monitors and on days not included in the training data for each fold). SHAP values describe the additive contribution to the prediction from every variable for each observation.

The SHAP overview plot illustrated different patterns of feature importance in Terra and Aqua (Fig. 5). The rank of the mean absolute SHAP values suggested that the top key contributing variables to predicting the magnitude of ΔCWV in the Terra dataset were time trend (even though all of the data in the testing set were from days not included in the training data, there was still clear seasonality when plotting the SHAP estimates), the magnitude of the MAIAC CWV itself, the blue band uncertainty estimate from MAIAC, the MAIAC AOD, the distance to the

nearest major water body, and the elevation. For the Aqua dataset, the blue band uncertainty ranked at the top, followed by the MAIAC AOD, the MAIAC CWV, the proportion of developed area in a 1x1 km square, time trend, the proportion of forest coverage in a 1x1 km square, and the relative azimuth angle. The SHAP values ranged from -0.52 to 0.82 cm for Aqua, and from -0.55 to 0.30 cm for Terra, aligning with the higher overall error in the Terra dataset.


For Terra, predicted ΔCWV values became larger in more recent years (Fig. 6.a). This suggests the observed positive bias has been getting stronger since ~2010. This trend was not observed in Aqua for which the time trend was a much weaker predictor. Similarly, a higher MAIAC CWV was also more likely to generate higher ΔCWV in Terra (a positive bias), and the influence was getting stronger along the time trend (Fig. 6.b). In contrast, in Aqua the

model suggested that MAIAC CWV conservatively underestimated extreme values in both seasons, although the overall impact was weaker (SHAP values closer to zero) and more stationary across time compared to Terra.

The impact of the rest of the features was similar for both Terra and Aqua (Fig. 7). Some outlying large AOD values had negative effects on the ΔCWV. Larger blue band uncertainty, higher elevation, or relative azimuth angle around

45 and 145 degrees increased the error. The SHAP estimates of global feature and individual datum contributions clearly diagnoses two main factors: 1) changing calibration of MODIS Terra NIR bands at 940nm over time, resulting in a trend of CWV bias from Terra, and 2) growing underestimation of CWV with increase in AOD. MAIAC CWV retrieval neglects the effect of aerosol scattering, which increases the measured radiances and the band ratios, resulting in underestimation of CWV.

**4.4 Prediction with new data**

To predict into a new dataset, we refit our XGBoost models by again running our nested random hyperparameter tuning using DART tree dropout, this time on the entire training dataset. For both models fit to the Aqua and Terra datasets, the optimal set of hyperparameters (selected from the same set of 50 candidates) was the same, including both L1 and L2 regularization (alpha and lambda), the deepest trees we permitted (maximum depth of 9), and no

more dropout (rate drop of 0) than the minimal random selection of one tree per model that had been fixed a priori (with the one drop option) (Table 3).





The resulting trained algorithm can generate ~3 million MAIAC CWV measurement error estimates per minute (on 4 cores) in new locations using the XGBoost predict function and these can be subtracted from the MAIAC CWV

value to generate a corrected CWV estimate for downstream use.

**4.5 Validation with SuomiNet GPS CWV**

As an external validation, we applied our XGBoost models to MAIAC data in 1 km * 1 km grid cells containing SuomiNet GPS stations. We removed about 20 observations (0.1% of the merged datasets) with outlying CWV values above 9 cm which were almost all from SuomiNet site P776 in year 2011. The resulting validation dataset

with collocated Terra or Aqua MAIAC CWV and SuomiNet CWV included 17,469 and 16,466 day-observations respectively from 57 SuomiNet stations (from years 2005 to 2015). SuomiNet CWV in the Terra collocated dataset had a mean of $1.57 \pm 1.04$ cm, while the Aqua collocated dataset had a mean of $1.50 \pm 1.01$ cm. The SuomiNet CWV had a more right-skewed distribution than MAIAC CWV.

After applying our correction, the MAIAC CWV had lower RMSE versus SuomiNet CWV compared with the raw MAIAC CWV in 53/57 sites for Terra and 56/57 sites for Aqua. The RMSE for agreement with SuomiNet CWV in the full validation dataset improved by 19.7% from 0.28 to 0.22 cm for Terra, and by 9.5% from 0.25 to 0.23 cm for Aqua. The Pearson's correlations of MAIAC CWV with SuomiNet CWV were improved by 1 percentage point from 0.969 to 0.978 for the Terra collocated dataset, and by 0.4 percentage points from 0.974 to 0.978 for the Aqua

dataset. Plotting the RMSE after correction at SuomiNet locations (grid cells where we make predictions), we observed higher RMSE (worse performance) near Lake Ontario and on the Atlantic coastline (Fig. 8.a). Most sites show improved RMSE after correction except 4 sites for Terra and 1 site for Aqua (Fig 8.b).

Another goal of addressing measurement error in satellite retrievals is to improve the comparability of different

instruments. Given changing atmospheric conditions within the same day between overpass times (Terra in late morning and Aqua in early afternoon), we use the corresponding two SuomiNet CWV measures to estimate the expected agreement. When restricting to days with both Terra and Aqua CWV observations collocated with SuomiNet observations, we had 9,940 station-days with all four measures. Raw MAIAC CWV had a Pearson correlation of 0.975 between Terra and Aqua, and after applying our correction this increased to 0.977, although this

was still slightly below that of the two corresponding within-day SuomiNet CWV measures with a correlation of 0.982. We demonstrate that our algorithmic correction slightly improves on the already excellent agreement of MAIAC CWV from Terra versus Aqua, but is still not quite as close as comparing pairs of within-day measures from the same ground instruments.

**5 Discussion**

The Northeastern USA exhibits large seasonal variation in CWV. While satellite retrievals using the MAIAC algorithm are overall excellent at estimating CWV, they also have seasonality in their measurement error versus





ground measurements from AERONET sun photometers. We show this measurement error has notable
heteroscedasticity (larger errors with greater CWV) and has been worsening, with time, for data derived from Terra.
Satellite retrievals using MODIS and similar platforms have considerable strengths for measurement of CWV based
on their global daily coverage and reconstruction of longer-term records during the satellite era. Our analysis
demonstrates that gradient boosting with XGBoost and features including satellite retrieval quality assurance,
aerosol optical depth estimates, land use terms, and time trends can substantially refine satellite-derived retrievals of
1 km * 1 km resolution CWV compared with sun photometer measures of CWV on test days and at sites that were
withheld from training data. Even with this rigorous cross-validation, our model explains 45.0% of the measurement
error from Terra CWV ($R^2$), and 24.1% of the measurement error in Aqua CWV. This is an impressive proportion of
the difference of MAIAC and AERONET CWV to explain given that the MAIAC CWV is already quite accurate
with a ΔCWV standard deviation of only 0.252 cm and 0.189 cm for Terra and Aqua respectively, in spite of
comparing a 1 km * 1 km satellite retrieval with point measurements from the AERONET sun photometers.

Strategies for model training and cross-validation of powerful algorithmic predictive methods need to reflect the
structure of the underlying data and the intended use of prediction models - otherwise overfitting may lead to an
inaccurate assessment of model performance. Given the sparsity of the collocated AERONET data, we decided to
assess performance in cross-validation that mimicked the desire to predict to new places (without AERONET
stations) and on dates without AERONET data (e.g. when sun photometers are out of service for recalibration).


While our XGBoost models are complex ensembles of one hundred boosted regression trees, we use the powerful
new SHAP method for interpretation of the importance of each variable and their contributions to individual
predictions. Contextualizing the magnitude of the SHAP value (for each variable) and examining the SHAP-based
contribution in visualizations along with the feature value distribution can also hint where retrieval algorithms can
be modified for better results. For example, although the measurement error was lower for Aqua, scatterplots for the
top two variables by SHAP suggest that MAIAC may underestimate CWV when the blue band uncertainty is very
low and may underestimate CWV at higher AOD values. For Terra, the date as an integer is the most important
feature, even though our cross-validation approach meant that all SHAP values were estimated for predictions made
on dates that did not occur in the training data. Based on the SHAP plots, the date predictor describes seasonal and
long-term trends related to an emerging positive bias for Terra that is worse in the summertime.

Demonstrating that there is an improvement in the agreement of corrected MAIAC CWV with the SuomiNet
measures is a strong validation for several reasons. First, the SuomiNet stations offer a well-validated measure of
CWV that relies on a different principle (tropospheric delay) from the sun photometry of the AERONET and the
MODIS satellite retrieval of the MAIAC algorithm. The second strength of this validation is that the SuomiNet
validation occurred at locations that are unique (not included in the training data from AERONET sites), including
many that are far away from the largely coastal AERONET stations in the Northeastern USA. Although Terra CWV
also had a larger measurement error versus SuomiNet CWV measures than Aqua CWV, after our correction using

XGBoost, the updated MAIAC CWV for Terra and Aqua both had lower RMSE values of 0.221 cm and 0.226 cm versus SuomiNet stations - suggesting that we may have achieved parity and perhaps reached the limits of this approach to correct for the sources of measurement error we considered in comparing this satellite retrieval product with point measures from ground stations.

Strengths of our empirical machine-learning approach include a fast algorithm that uses only a few variables, primarily already included in the MAIAC retrieval suite and derived land use terms, to correct measurement error in CWV. Limitations of using MODIS-derived CWV from MAIAC include the availability of few measurements per day (versus geostationary satellites) and restriction to cloud-free and daytime values. Our measurement error model has not yet been evaluated for how well it would have worked in a region with substantially fewer AERONET stations or very different climate conditions.

**6 Conclusions**

Empirically correcting for measurement error with machine-learning algorithms is a relatively easy post-processing opportunity to improve satellite-derived CWV data quality for Earth science and remote sensing applications. Furthermore, the use of machine-learning interpretation tools points to potential sources of measurement error (e.g. a positive bias in CWV retrievals from Terra that is worse in more recent years) that can help when refining satellite 370 retrieval strategies. We demonstrate that a parsimonious nine-predictor XGBoost model for updating satellite-based column water vapor from the MAIAC retrieval based on AERONET values can decrease measurement error as validated at an independent network of ground sensors across the North Eastern USA.

**Author contribution**

AJ designed the study and supervised analysis. YL carried out the analysis including developing code and running 375 the models. YW and AL provided MAIAC data and valuable guidance. JR, MD and IK created the analytic datasets. RC and MSH provided guidance on remote sensing principal approaches. AJ and MSH prepared the manuscript with contributions from all co-authors.

The authors declare that they have no conflict of interest.

**Acknowledgments**

We thank Kodi B. Arfer for assistance with the XGBoost hyperparameter tuning implementation. We thank the AERONET federation PIs and their staff for establishing and maintaining the 75 sun photometer sites used in this investigation available from https://aeronet.gsfc.nasa.gov/. We thank the UCAR and the SuomiNet program for the



university-based GPS network data available at https://www.suominet.ucar.edu/. MAIAC data were downloaded from ftp://dataportal.nccs.nasa.gov/DataRelease on 2016-10-16. Elevation data was obtained from the National Elevation Dataset at https://catalog.data.gov/dataset/usgs-national-elevation-dataset-ned. Land use data were obtained from the National Land Cover Database 2011 at https://www.mrlc.gov/.

All data and code to reproduce the analyses in this study are available and archived at

https://doi.org/10.5281/zenodo.3568449

This research was funded by NIH grants UH3 OD023337 and P30 ES023515 and grant 2017277 from the Binational Science Foundation. A.C.J. was supported by NIH grant R00 ES023450.

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



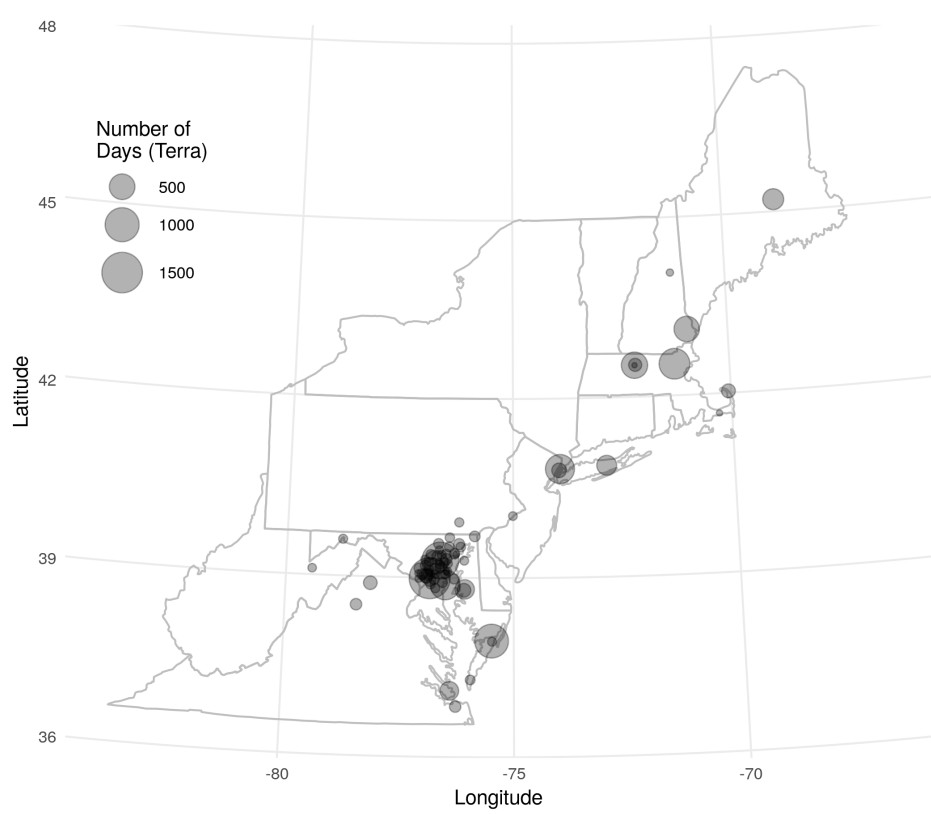


**Figure 1. Study region in Northeastern and Mid-Atlantic USA with 75 unique AERONET stations showing the number of days with observations from the collocated Terra dataset.**






Figure 2. Example of training (blue) and testing (brown) datasets of one fold in ten-by-ten-fold cross-validation. Prediction models are only evaluated on days and at stations that were not used in model training to avoid overfitting.

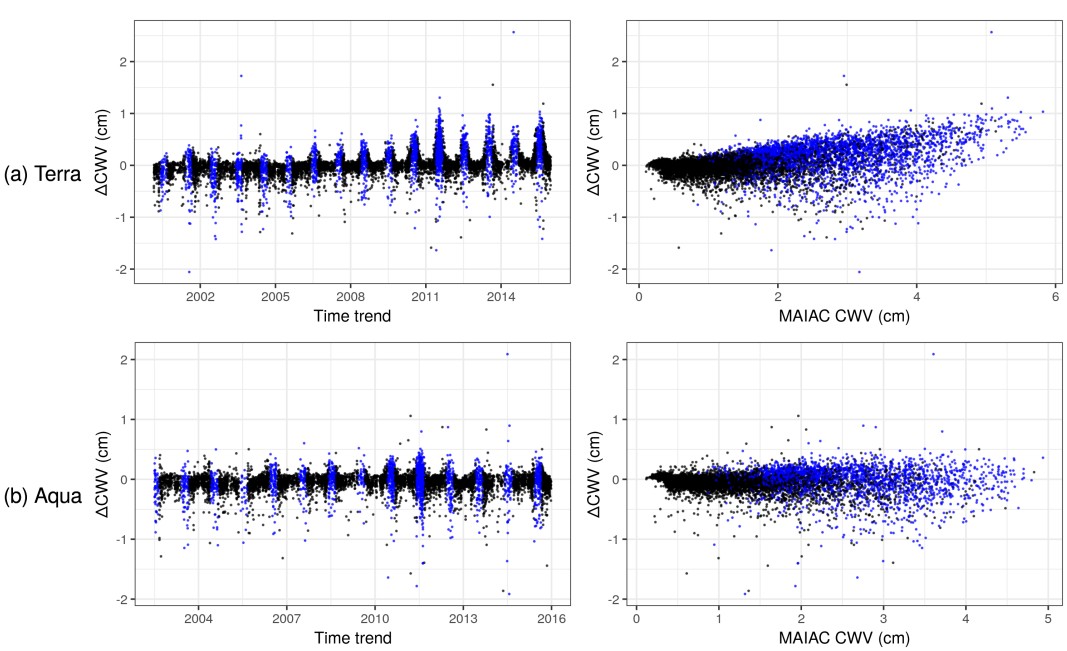

Figure 3. Scatterplots of ΔCWV versus time trend and MAIAC CWV in Terra (a) and Aqua (b). Observations in the summer months (June - August) are colored in blue.

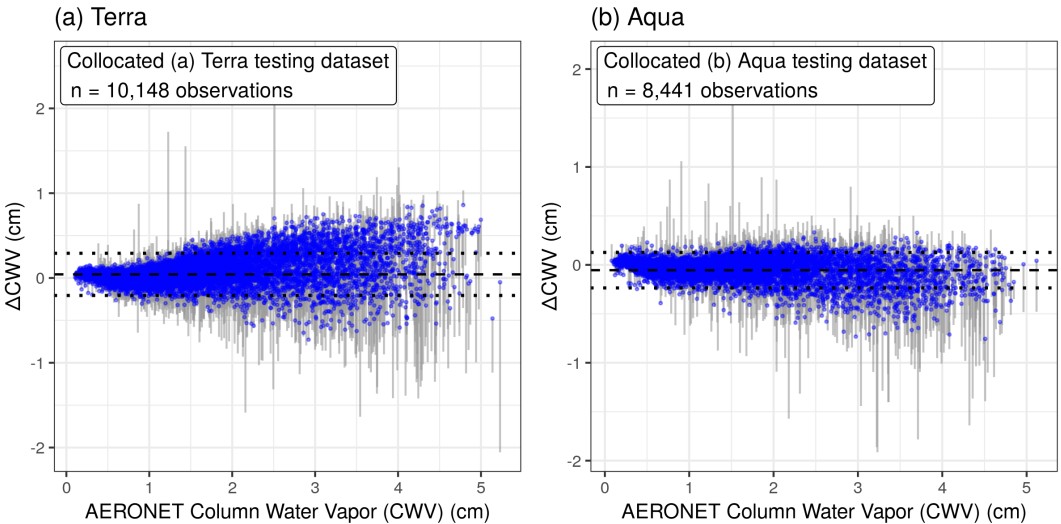

**Figure 4. The difference between MAIAC and AERONET CWV values (ΔCWV) was reduced in cross-validation of collocated (a) Terra and (b) Aqua data. The corrected values of ΔCWV are shown with blue points, segments connect back to the measurement error from the raw ΔCWV. The dotted lines show one standard deviation from the mean (the dashed line near zero).**

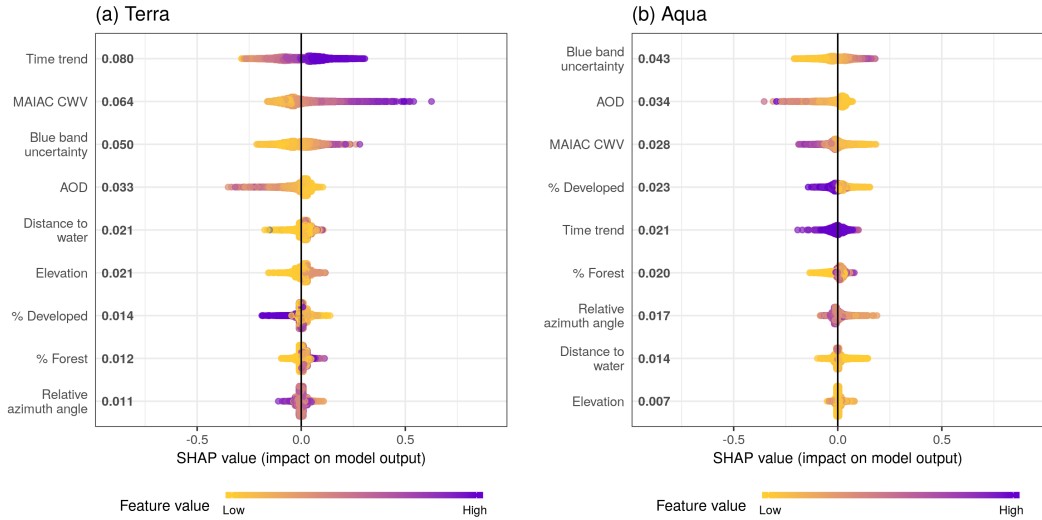

**Figure 5. Sina plots show the distribution of feature contributions to predictions of CWV measurement error using SHAP values of each feature for every observation. The x-axis is set between -1 and 1 to facilitate comparison across subpanels showing models for Terra and Aqua datasets. Features were ordered on the y-axis by their mean absolute SHAP values**





over all observations (bold on the right of the variable names, units are the same as ΔCWV predictions in cm). The color is scaled to the feature value (purple high, yellow low).


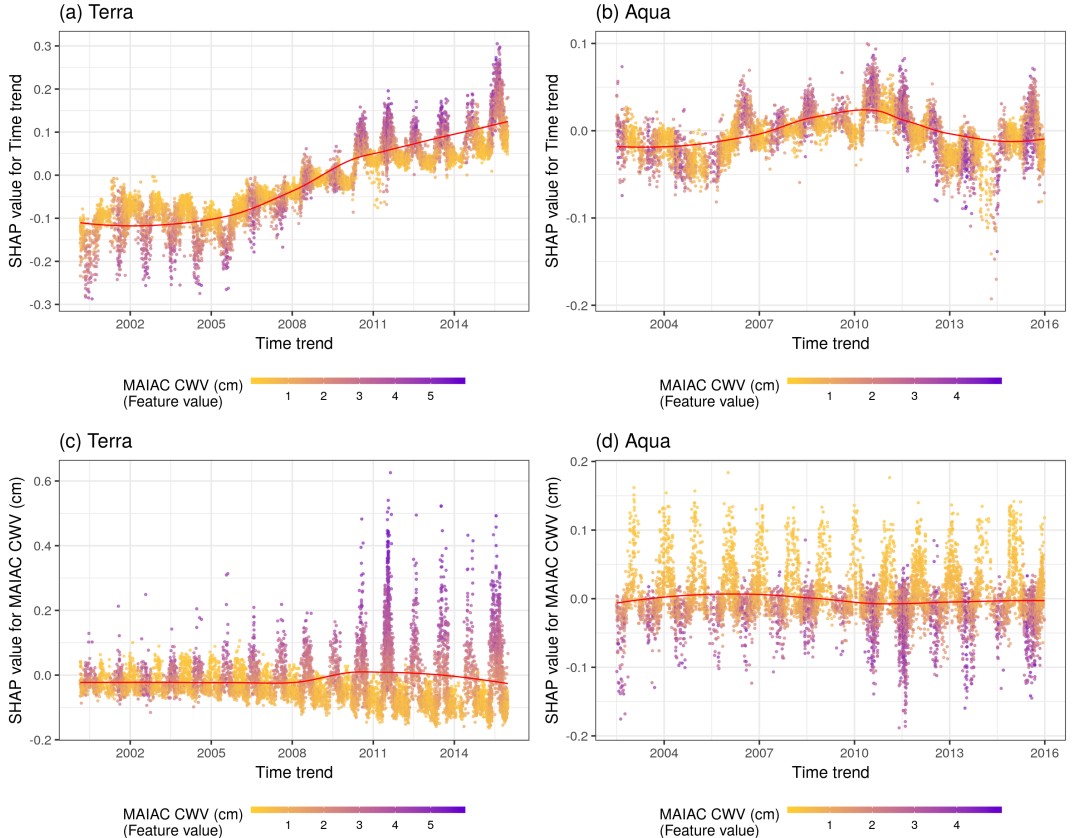

**Figure 6.** SHAP values showing the contribution of the time trend to predictions for Terra (a) and Aqua (b). The color represents the MAIAC CWV for each observation (purple high, yellow low). The LOESS (locally estimated scatterplot smoothing) curve is overlaid in red. Terra (c) and Aqua (d) SHAP values showing the contribution of the MAIAC CWV to predictions of CWV measurement error shown across the time period of the study. Note distinct y-axis scales for Terra and Aqua datasets. The color represents the MAIAC CWV for each observation (purple high, yellow low).


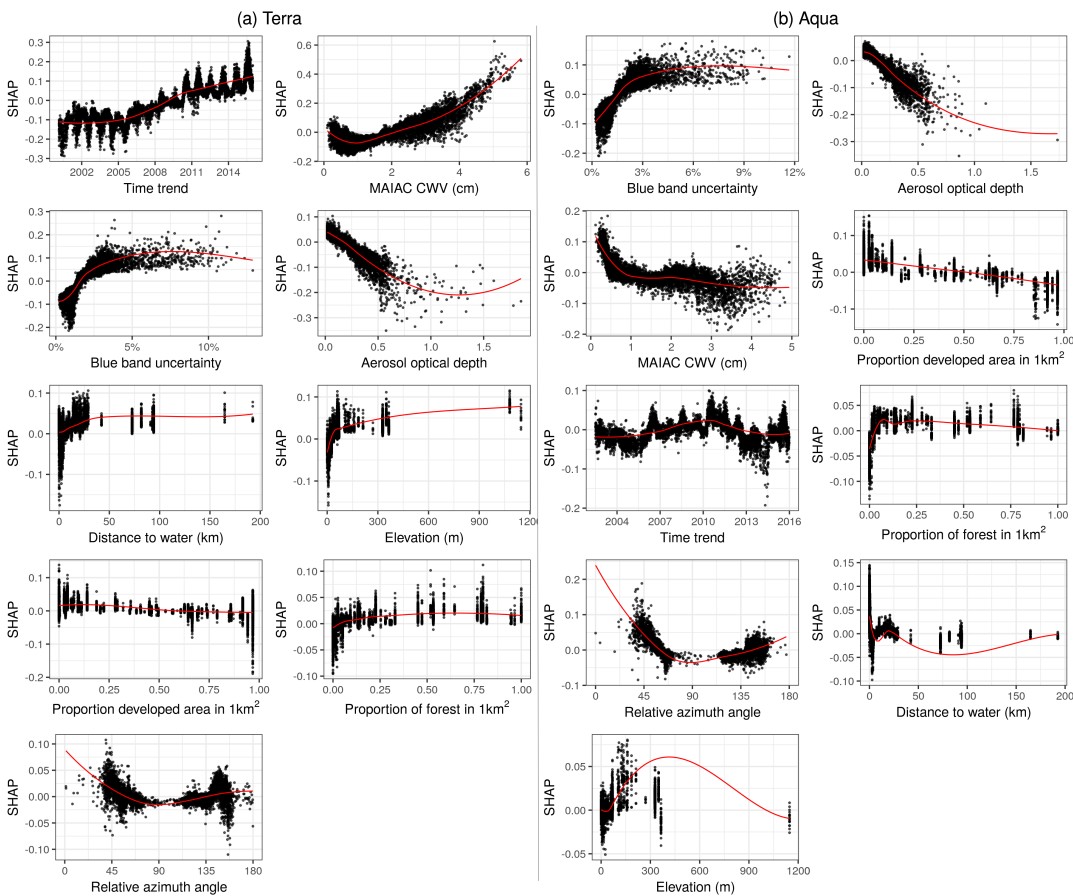

**Figure 7. Descriptive scatterplots of the features versus their SHAP scores approximating their contribution to the**
**predictions for ΔCWV (cm) on the y-axis. Subplots are ordered by overall variable importance (mean absolute SHAP**
**score, see Fig. 5) by satellite.**



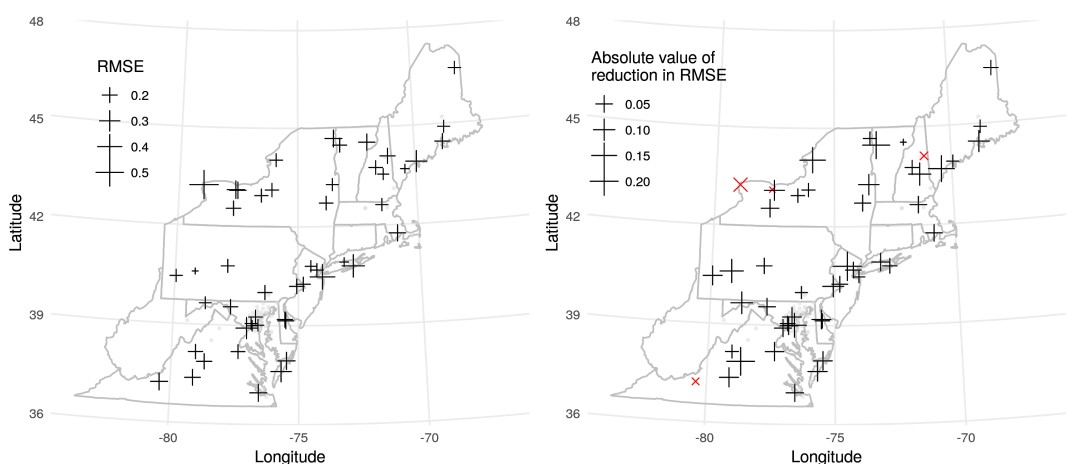

**Figure 8. (a) RMSE between algorithmically-corrected Terra MAIAC CWV and GPS-based CWV for each SuomiNet**
**station shown as crosses. AERONET sites used to train the model are shown as points. (b) the difference in RMSE versus**
**GPS CWV using the corrected MAIAC CWV relative to using the original MAIAC CWV by SuomiNet station. The four**
**sites (out of 57 total) having higher (worse) RMSE after correction are shown with red X symbols.**





**Table 1. Descriptive Statistics of MAIAC, AERONET CWV and the ΔCWV for Terra and Aqua by Season.**

| Terra Mean (SD) (cm) | Spring (N = 2,265) | Summer (N = 3,150) | Fall (N = 3,101) | Winter (N = 1,632) | Total (N = 10,148) |
|---|---|---|---|---|---|
| MAIAC CWV | 1.18 ± 0.82 | 2.71 ± 0.94 | 1.39 ± 0.82 | 0.56 ± 0.30 | 1.62 ± 1.12 |
| Aeronet CWV | 1.20 ± 0.80 | 2.54 ± 0.90 | 1.40 ± 0.79 | 0.60 ± 0.34 | 1.58 ± 1.04 |
| ΔCWV | -0.014 ± 0.192 | 0.172 ± 0.322 | -0.009 ± 0.202 | -0.032 ± 0.095 | 0.043 ± 0.249 |

| Aqua Mean (SD) (cm) | Spring (N = 1,921) | Summer (N = 2,276) | Fall (N = 2,715) | Winter (N = 1,529) | Total (N = 8,441) |
|---|---|---|---|---|---|
| MAIAC CWV | 1.12 ± 0.74 | 2.49 ± 0.84 | 1.33 ± 0.73 | 0.55 ± 0.27 | 1.45 ± 0.98 |
| Aeronet CWV | 1.16 ± 0.78 | 2.52 ± 0.90 | 1.42 ± 0.78 | 0.60 ± 0.33 | 1.51 ± 1.01 |
| ΔCWV | -0.049 ± 0.150 | -0.027 ± 0.253 | -0.086 ± 0.159 | -0.047 ± 0.101 | -0.054 ± 0.181 |

*Note.* Means and standard deviations (units of cm) are shown for three-month seasons (Spring: MAM; Summer: JJA; Fall: SON; Winter: DJF) across all the years and the total.



**Table 2. Predictive Performance in the Testing Dataset Comparing Three Cross-validation Strategies**

|  | Terra dataset | Aqua dataset |
|---|---|---|
| Overall variation | SD 0.25 cm | SD 0.19 cm |
| Split by day (5 fold) | RMSE 0.15 (57.8%) $R^2$ = 65.6% | RMSE 0.14 (76.3%) $R^2$ = 36.5% |
| Split by station (5 fold) | RMSE 0.18 (71.4%) $R^2$ = 47.5% | RMSE 0.16 (83.8%) $R^2$ = 23.5% |
| Split by station and day (10*10 fold) | RMSE 0.18 (73.1%) $R^2$ = 45.0% | RMSE 0.16 (83.5%) $R^2$ = 24.1% |

*Note.* The relative percentage of RMSE compared to overall variation (SD) is listed beside RMSE.



**Table 3. Hyperparameters for the Fully Trained XGBoost Model**

| | Selected values |
| --- | --- |
| eta | 0.44 |
| max_depth | 9 |
| gamma | 0.099 |
| lambda | 38 |
| alpha | 0.0023 |
| rate_drop | 0 |
| one_drop (fixed a priori) | True |
| nrounds (fixed a priori) | 100 |
