# Peer review of "Gradient Boosting Machine Learning to Improve Satellite-Derived Column Water Vapor Measurement Error"

_Atmospheric Measurement Techniques, 2019_

## Referee Comment (RC1) · Anonymous Referee #2 · 29 Jan 2020

The Authors propose a machine learning method to perform an a posteriori correction of MODIS water vapour retrievals produced by the MAIAC algorithm. Based on a training dataset of MODIS-AERONET coincidences, they identify a number of variables from which differences between MODIS and AERONET water vapour estimates can be predicted and they train a model to correct for such differences. MAIAC retrievals post-processed with the trained algorithm are then compared to SuomiNet water vapour data, and a reduction in the MAIAC-SuomiNet RMS difference is observed compared to the raw MAIAC retrievals.

During the access review I asked the Authors to somewhat expand their explanation

of the algorithm they adopt, and the Authors have accommodated my request to some extent. However, the description of the methodology still looks vague in some respects, and does not allow to easily understand which computational steps are performed in order to attain the presented results. The discussion still uses a number of terms without defining them and takes for granted concepts that do not look particularly obvious. Apart from that, the results look good, and the paper may be published after the Authors account for the following comments.

- L64-65. What do you mean by "weak predictor"? And "binary partitioning" of what?

- L113-122. The problem with this paragraph is similar to the one I highlighted during the access review, namely that it uses many specialized terms without defining them. Therefore, this paragraph is not very informative to a reader who does not have a background in this type of methods (a condition that is probably not uncommon among the readership of AMT), and probably is also not very informative to a reader who does. In particular, the following aspects are not clear to me.

1. Let us suppose that we have a number of predictors (e.g. solar zenith angle, viewing zenith angle, AOD, etc.). When you say that the model "specifies a few recursive binary splits of predictors etc.", do you mean that it defines a threshold for each predictor and returns a different output depending on whether the predictor is above or below the threshold? And does the next level of the tree apply similar operations to the result of this first thresholding, and so on? If so, make this point clearer in your discussion. I had to look inside the references to understand this, but such a basic level of detail should be already understandable from your paper, without forcing the reader to peruse the references.

2. Who decides which predictors should be split and whether they should be split independently or according to certain logical combinations (AND, OR, etc.)? Is it the user or is it the training algorithm that makes this decision? In addition, if this is up to the training algorithm, how is the system trained? How is the cost function defined and

how are the system parameters adjusted?

3. Again, what do you mean by "weak learner"? How are multiple learners combined? Who decides what weight should be given to each learner, and how?

4. What is the role of "gradient" in gradient boosting? Gradient of what with respect to what?

- L130. Please define the "several hyperparameters related to the desired size and complexity of the model". Plus, why "hyperparameters" and not simply "parameters"?

- L132. What do you mean by "nested comparison"? In particular, in what sense "nested"?

- L133. Could you provide a reference for Latin hypercube sampling, and possibly summarize what it essentially does?

- In general, the fundamental question I have about Section 3 is: if I want to replicate your study or apply your method to another problem - e.g., by writing my own code - what do I actually need to do? What are the computational steps involved?

- L146. I think "Shapely" should actually read "Shapley"

- L442. Some details of the reference appear to be missing.

––––––––––––––––––––

---

## Referee Comment (RC2) · Anonymous Referee #3 · 26 Feb 2020

General

The manuscript is appears smartly prepared and well written. However, to my opinion the manuscript is not appropriate for publication in AMT. The manuscript deals with a machine learning concept to improve the MODIS/MAIAC column water vapour retrieval. Only machine learning aspects are discussed, and these aspects are just described. There is no chance for the reviewer to check the quality of the work. I have to believe what they write. This is rather unsatisfactory.

MAIAC needs a large bunch of surface, atmospheric, and technical input parameters to successfully retrieve water vapour information. These input parameters have partly

large uncertainties. The surface and atmospheric input data change from day to day, with time (morning vs afternoon), with season, with land use changes. Nevertheless, the MAIAC methodology seems to be very robust, the accuracy of the MAIAC products is very good (without any machine learning effort)! To my opinion, it is impossible to further improve the MAIAC column water vapour values! However, the authors of the manuscript want to convince the reader that the machine learning concept overcomes this insurmountable wall of given and (unknown) uncertainties. It improves the results, and reduces the overall uncertainties although the given uncertainties are unknown! How is that possible? The paper gives not answer to this.

I recommend a rejection also because the paper does not contain any aspect of atmospheric science or technological development that would justify publication in AMT. Only machine learning aspects are given. Why not choosing again Remote Sensing where the first paper of Just et al. was published?

Some details:

The title is 'strange' , not logical! What does it mean: . . . to improve . . .. the error. . .? What does it mean: . . . satellite-derived . . . measurement??? The column water vapour is clearly a retrieval product. . .. There is no 'direct' measurement.

Lines 85-90: In the introduction it is written: machine learning approaches such as XGBoost can model complex phenomena etc.. . . The resulting prediction model can provide an algorithm to reduce the retrieval errors. I conclude: yes, the model can do that provided the complex input parameter set is free of uncertainties. But many aspects (input data) are not well known in the case of the MAIAC retrieval, uncertainties in the input data are large and that is the reason for the uncertainties in the product.

Section 2:

Line 118-120: Target modelling parameter is the difference between MAIAC and AERONET CWV. . . My question is: When the machine learning approach finds the

best way for correction (e.g. based on all the 75 station of northeastern United States in Figure 1) can this approach then be applied to the rest of the world? I do not believe that this will work! Probably we have to find optimum ways for corrections again and again, region by region and all this for different seasons.

Section 3

Some examples that explain my general feeling with the paper: Lines 147-149: The XGBoost package is used! Ok! But the reference for this is a conference contribution, grey literature!

Lines 153 154: XGBoost is combined with DART (here the reference does not indicate any journal?). Can we believe, everything is ok with this procedure? Can we trust? Is all the material peer reviewed by machine learning experts?

Lines 159-161: Bayesian optimization for hyperparameter tuning of XGBoost models was performed using the autoxgboost R package (Thomas et al, 2018). The reference points to arXiv…. This is a preprint archive (no peer review, nothing). So, what is this? Can we trust?

Lines 175-176… The contribution of each feature to cross-validated predictions was estimated by SHAP values (reference… arXiv). …again this preprint archive…

Lines 177-227: A lot of information and description is given by the authors, written in a smart appearing way, but it does not help. The reader is lost! He/she just has to believe that everything is ok with this way. But he/she does not trust.

Section 4: results:

I avoid to give my comments to the text… nobody can check what they state…, what is ok, what is not ok, what is trustworthy, what is not trustworthy. There is nothing to judge!

To the figures…

Figure 1: There is no hint where we are? no city name, e.g., Boston, New York, no name of any state... Maryland.... Figure 1 is a nice 'indicator' , .... of the feeling I have with the entire paper.

Figure 3 and the following figures tell me: MAIAC does a good job, seasonally dependent uncertainties are visible. This is ok, surface properties change and are not perfectly considered in the retrieval. One should accept that.

Machine learning procedures may purge the deviations in this specific 'learning region' of Northeast USA. But for any new region ....? We have to start again, I believe.

Conclusion section: just 9 lines: This is good, there is practically nothing to conclude.

––––––––––––––––––––––––––––––

---

## Author Response (AR1)

Dear Editor:

As requested, we have responded to the reviewers' comments point-by-point. We have also made many responsive revisions to our manuscript which is substantially improved and includes greater clarity and detail on the machine-learning methods that we demonstrate in our empirical results. We would also like to point out that two previous referees stated in the access review that they felt the manuscript was an appropriate fit for inclusion in AMT. We have responded point-by-point and presented evidence of our scientific rigor in our data analysis. Finally, the ultimate test of the replicability of our empirical results lies in our use of the Open Access archival repository Zenodo where we have placed an irrevocable copy of all of our data and R code to regenerate all of our results. We thank the editor for their consideration of our work and for the opportunity to improve our manuscript through the AMT discussion process.
* * *
Response to RC1: We thank the reviewer for their helpful and specific comments. We completely agree that revisions contributing to more accessible methods and discussion sections will increase the impact of our work. We have added plain language descriptions that explain the meaning behind specialized terms that are commonly used in machine learning. More importantly, our more accessible language is to be taken in conjunction with the reproducible data + open source R code we have archived in the Zenodo open-access digital repository ((https://doi.org/10.5281/zenodo.3568449), which will ease the integration of the methods and ideas that we employ into other atmospheric measurement applications. We have substantively revised the language we use to describe methods within the introduction, methods, and discussion sections. We add additional details, define terms coming from the field of machine learning, and seek to clarify the points of confusion raised by the reviewer. We have also included references to two strong introductory articles to guide readers that are seeking additional information on the inner workings of the machine-learning methods that we describe. We hope that the referee also sees the improved clarity and accessibility of the revised manuscript which we consider substantially improved. We include a point-by-point response to comments below:

> 1. - L64-65. What do you mean by "weak predictor"? And "binary partitioning" of what?
>
> > Response: We have added additional information to clarify this sentence in our introduction which previously read, "Gradient boosting involves fitting a large number of tree-based models where each subsequent tree is made a weak predictor of the error from the previous trees." The revised section now reads, "XGBoost involves fitting a large number of tree-based models. Each subsequent tree is fit to the error from the previous trees and the predictions of all the trees are added together. Each tree's prediction is multiplied by a shrinkage factor (or "learning rate") η, a number between 0 and 1. By adding successive trees, XGBoost descends the gradient of the loss function. The component trees use a recursive binary partitioning of the predictors that accommodates varying types and scales of predictor variables and is

robust to outliers (Elith et al., 2008)." To further clarify, both the shrinkage factor η and the use of random dropout (we explain the DART method elsewhere), are used to decrease overfitting that occurs when a model is directly applied to residual error. Our revisions have clarified that the binary partitioning used in regression trees is a splitting of the predictor variables.

*2. - L113-122. The problem with this paragraph is similar to the one I highlighted during the access review, namely that it uses many specialized terms without defining them. Therefore, this paragraph is not very informative to a reader who does not have a background in this type of methods (a condition that is probably not uncommon among the readership of AMT), and probably is also not very informative to a reader who does.*

Response: We have added additional detail to explain specialized terms and have added a reference to an accessible introduction to regression trees. Below we include specific examples of how we have clarified points that were not clear to the reviewer.

*3. In particular, the following aspects are not clear to me. Let us suppose that we have a number of predictors (e.g. solar zenith angle, viewing zenith angle, AOD, etc.). When you say that the model "specifies a few recursive binary splits of predictors etc.", do you mean that it defines a threshold for each predictor and returns a different output depending on whether the predictor is above or below the threshold? And does the next level of the tree apply similar operations to the result of this first thresholding, and so on? If so, make this point clearer in your discussion. I had to look inside the references to understand this, but such a basic level of detail should be already understandable from your paper, without forcing the reader to peruse the references.*

Response: Following the reviewer's suggestion, we have revised this section to explain in greater detail how tree-based regression models operate. While avoiding jargon, we also see our paper as an opportunity to inform a new readership on the terminology of machine learning and how it can be applied in earth sciences. It now reads: "For an introduction to regression trees, see Strobl et al. (2009). A regression tree is a model that specifies recursive binary splits of predictors and assigns a constant value to all cases that end up in the same terminal node (namely, their mean on the dependent variable). The algorithm chooses the splits across all predictors that minimize the variance of the residuals. The maximum number of splits within each tree (also known as the maximum depth) can be set as a hyperparameter. A set of multiple trees can be used for prediction by combining the outputs of the individual trees for each case. Such a set of trees can accommodate complex relationships including non-linearities and interactions while being robust to outliers. Boosting is a method of fitting a series of models iteratively, with each model fit on the residuals of the previous models. While each tree may individually perform relatively poorly at predicting the outcome (and thus is known as a "weak learner"), the combination of many trees can collectively describe complex relationships and account for the impact of many predictors. Further, because boosting includes

sequentially learning by combining many iteratively fit trees that address the error in previous trees, this technique performs well, achieving low testing error. The XGBoost package is a scalable gradient boosting implementation with additional features including penalties to avoid overfitting and optimized computational speed (Chen and Guestrin, 2016)." We believe our revisions achieve a reasonable balance of including sufficient descriptive information and useful references without swamping the reader with excessive detail.

*4. Who decides which predictors should be split and whether they should be split independently or according to certain logical combinations (AND, OR, etc.)? Is it the user or is it the training algorithm that makes this decision? In addition, if this is up to the training algorithm, how is the system trained? How is the cost function defined and how are the system parameters adjusted?*

Response: As clarified in our revised Statistical Methods section, the regression tree algorithm selects at each step the split across all predictors that minimizes the variance of the residuals. The selection of splits is done recursively and the number of splits (depth of the tree) is a hyperparameter that is tuned by the analyst (we discuss our hyperparameter tuning approach below). While a split upon another split of a tree constructs a logical AND statement, the addition of sufficient splits can approximate an OR statement. The user does not select split points.

*5. Again, what do you mean by "weak learner"? How are multiple learners combined? Who decides what weight should be given to each learner, and how?*

Response: the "weak learner" term was explained in the revised manuscript. It now reads: "While each tree may individually perform relatively poorly at predicting the outcome (and thus is known as a "weak learner"), the combination of many trees can collectively describe complex relationships and account for the impact of many predictors. "

Instead of fitting a single large decision tree to all the data with many splits, which will perform poorly in prediction (having low bias but very high variance), the boosting approach learns slowly by fitting a smaller decision tree to the residuals of the model and slowly improving the model in areas where it does not perform well. In general, statistical learning approaches that learn slowly tend to perform well by producing both low bias and low variance.

*6. What is the role of "gradient" in gradient boosting? Gradient of what with respect to what?*

Response: The gradient in question is the residuals (the observed values minus the predicted values), which are the gradient of squared-error loss with respect to the residuals. A gradient-boosting model adds trees in order to minimize this gradient.

*7. - L130. Please define the "several hyperparameters related to the desired size and complexity of the model". Plus, why "hyperparameters" and not simply "parameters"?*

Response: In machine learning, a hyperparameter is a configuration set before the learning process begins and that takes values that cannot be directly estimated from the data. Parameters, such as regression coefficients or split points in tree-based models, are estimated from the data. Simple algorithms like linear regression don't require hyperparameters while more complex algorithms may have several hyperparameters. For these more complex machine-learning algorithms, hyperparameters need to be predefined by researchers within a certain range of values. Often a set of appropriate hyperparameter values that result in improved performance are selected through a cross-validation process. We have added the following sentences to clarify, "XGBoost has several hyperparameters related to the desired size and complexity of the model that need to be set in training for each dataset. We had *a priori* selected to tune our XGBoost models with DART using six hyperparameters (Supplementary Table S2), while using default values for other potential hyperparameters based on previous modelling experience."

8. - L132. *What do you mean by "nested comparison"? In particular, in what sense "nested"?*

Response: To avoid overfitting, it is important that all learning, including the selection of hyperparameter values, occurs within the training dataset. However, the evaluation of performance should still be done on data that was not used in algorithm training and thus requires cross-validation within the training dataset. This leads to nested cross-validation, where the training data is being further split in half in order to evaluate the performance of different hyperparameter values. Our revised section now reads, "Our tuning and evaluation approach used two-level (nested) cross-validation. Within each training fold for our outer cross-validation, we further randomly split the training data in half and performed a 2-fold cross-validation to compare the performance of XGBoost models using 50 random sets of potential hyperparameters selected with Latin hypercube sampling (Stein, 1987) to be well-spaced across the range of potential hyperparameter values."

9. - L133. *Could you provide a reference for Latin hypercube sampling, and possibly summarize what it essentially does?*

Response: We use Latin Hypercube Sampling (LHS) to generate random combinations of parameter values. It is based on the Latin square design, which has a single sample in each row and column. One-dimensional LHS involves dividing your cumulative density function into n equal partitions; and then choosing a random data point in each partition. We are using a multi-dimensional LHS because we have six hyperparameters to tune across simultaneously. LHS helps to ensure that samples are representative of the real variability in the distribution. We have added additional explanation of the purpose of the Latin hypercube sampling and our section now reads, "Within each training fold for our outer cross-validation, we further randomly split the training data in half and performed a 2-fold cross-validation to compare the performance of XGBoost models using 50 random sets of potential hyperparameters selected with Latin hypercube sampling (Stein, 1987) to be wellspaced across the range of potential hyperparameter values. While this is more similar to a random search than a grid search, it is expected to more efficiently find well performing sets of hyperparameters than random search, because it decreases the likelihood of checking combinations that are trivially different or leaving unexplored regions in the six-dimensional space, which has too many combinations to effectively cover with a grid search. We selected the set of hyperparameters that minimized the RMSE within the withheld portion of the training data before refitting with all training data."

10. - *In general, the fundamental question I have about Section 3 is: if I want to replicate your study or apply your method to another problem - e.g., by writing my own code - what do I actually need to do? What are the computational steps involved?*

Response: We have taken several steps to assist our readers with replication. We have added substantially to the detailed description of our methods, and as we have listed in our manuscript, we have already placed full reproducible R code, including all computational steps, and our datasets in an Open Access Zenodo repository (DOI 10.5281/zenodo.3266058) enabling anyone to rerun our full code and regenerate all of our results in the manuscript. This makes checking the quality and reproducibility of our work fully accessible and will assist readers in replicating this study in new datasets or in applying these methods to other problems in atmospheric measurement science.

11. - *L146. I think "Shapely" should actually read "Shapley"*

Response: we have corrected this in the revised manuscript.

12. - *L442. Some details of the reference appear to be missing.*

Response: we have corrected the reference which appeared to be missing journal information.
* * *
Response to RC2: We thank the reviewer for their comments and the opportunity for us to frame more clearly why our work will be a contribution to AMT. We have responded point-by-point to the comments below.

1. *Comment: The manuscript deals with a machine learning concept to improve the MODIS/MAIAC column water vapour retrieval. Only machine learning aspects are discussed, and these aspects are just described. There is no chance for the reviewer to check the quality of the work. I have to believe what they write. This is rather unsatisfactory.*

Response: We have worked with care and used rigorous scientific methods in our data analysis. We have added substantially to the detailed description of our methods (see also our response to Referee 1), and as we have listed in our manuscript, we have already placed full reproducible code and our datasets in the Open Access Zenodo repository (DOI 10.5281/zenodo.3266058) enabling anyone to rerun our full code and regenerate all of our results in the manuscript. This makes checking the quality and reproducibility of our work fully accessible. The contribution of machine-learning models to improving column water vapor retrievals is the main point of our manuscript and we present in our manuscript both a description of our approach and empirical results at AERONET stations and an independent validation dataset from SuomiNet.

*2. Comment: MAIAC needs a large bunch of surface, atmospheric, and technical input parameters to successfully retrieve water vapour information. These input parameters have partly large uncertainties. The surface and atmospheric input data change from day to day, with time (morning vs afternoon), with season, with land use changes. Nevertheless, the MAIAC methodology seems to be very robust, the accuracy of the MAIAC products is very good (without any machine learning effort)! To my opinion, it is impossible to further improve the MAIAC column water vapour values!*

Response: We wholeheartedly agree that the MAIAC suite of products are very good. However, in collaboration with the MAIAC PI (our co-author on this paper, Dr. Alexei Lyapustin), we have worked to identify opportunities to further understand and reduce retrieval error in the MAIAC column water vapor product. While the recently published global validation of the MAIAC column water vapor product (Martins et al. *Atmos Res*. 2019) has noted the temporal drift in Terra CWV records, ours is the first analysis that demonstrates an empirical correction. Furthermore, our machine-learning model accounts for the complex interactions of input parameters rather than considering each one separately. Our method clearly demonstrates an improvement in the MAIAC column water vapor values.

*3. Comment: However, the authors of the manuscript want to convince the reader that the machine learning concept overcomes this insurmountable wall of given and (unknown) uncertainties. It improves the results, and reduces the overall uncertainties although the given uncertainties are unknown! How is that possible? The paper gives no answer to this.*

Response: Our empirical method to update the CWV values does not rely on propagation of estimated uncertainties for each of the inputs to the retrieval algorithm. Instead, we estimate the retrieval error versus AERONET stations and then build a statistical model to explain this retrieval error using a pre-defined set of input variables. Although we do not know the uncertainties in the retrieval parameters, our use of the new SHAP method for explainable machine-learning helps to quantify and rank which of the input variables we considered are the largest contributors to the retrieval error that we estimated.

*4. Comment: The title is 'strange' , not logical! What does it mean: . . . to improve . . .. the error. . .? What does it mean: . . . satellite-derived . . . measurement??? The column water vapour is clearly a retrieval product. . .. There is no 'direct' measurement.*

Response: In recognition of the referee's concerns we have modified our title to be: "Gradient Boosting Machine Learning to Reduce Satellite-Derived Column Water Vapor Retrieval Error"

*5. Comment: Lines 85-90: In the introduction it is written: machine learning approaches such as XGBoost can model complex phenomena etc.. . . The resulting prediction model can provide an algorithm to reduce the retrieval errors. I conclude: yes, the model can do that provided the complex input parameter set is free of uncertainties. But many aspects (input data) are not well known in the case of the MAIAC retrieval, uncertainties in the input data are large and that is the reason for the uncertainties in the product.*

Response: Our empirical results demonstrate that we are able to reduce retrieval error in the MAIAC product even without knowing the degree of uncertainty in the individual input datasets. Our strategy of modeling the difference between MAIAC retrievals and a ground truth observation works because there are informative predictors that explain much of the retrieval error, whether these predictors are directly related to the source of uncertainty or are themselves proxies (such as time trend). We have a long-running collaboration with co-author and MAIAC PI Dr. Alexei Lyapustin and a track record of using this approach to quantify and reduce retrieval error.

*6. Section 2: Line 118-120: Target modelling parameter is the difference between MAIAC and AERONET CWV. . . My question is: When the machine learning approach finds the best way for correction (e.g. based on all the 75 station of northeastern United States in Figure 1) can this approach then be applied to the rest of the world? I do not believe that this will work! Probably we have to find optimum ways for corrections again and again, region by region and all this for different seasons.*

Response: We agree with the reviewer that the generalizability of our specific model to new regions is untested and we have acknowledged this in our limitations. However, our approach (and reproducible code) may be applied in other regions with ground AERONET stations in future applications. Our results include all seasons and all years through 2015 of the MODIS record and our validation at SuomiNet stations shows that our results hold across the Northeastern USA, including at ground stations that are hundreds of kilometers from the nearest AERONET station used in training.

*7. Section 3 Some examples that explain my general feeling with the paper: Lines 147-149: The XGBoost package is used! Ok! But the reference for this is a conference contribution, grey literature!*

> Response: XGBoost is very widely used and has recently emerged as a leading tool often winning machine-learning competitions. We have added additional citations related to its performance but the convention in the rapidly evolving machine-learning discipline has included a greater use of competitive conference proceedings which are rigorously evaluated and empirically benchmarked through shared code (their field has thus largely avoided the overhead of working with major commercial publishers). As an example, Google Scholar lists 4,856 citations since 2016 for this XGBoost paper that we have cited (as of February 27, 2020). We would not consider this publication to be grey literature. The XGBoost software implementation is also widely used and is developed by a sophisticated community of open source programmers - the software repository at https://github.com/dmlc/xgboost, which has 392 contributors as of February 28, 2020.

*8. Lines 153 154: XGBoost is combined with DART (here the reference does not indicate any journal?). Can we believe, everything is ok with this procedure? Can we trust? Is all the material peer reviewed by machine learning experts?*

> Response: We thank both referees for drawing our attention to an incomplete citation. We have corrected our reference for the DART manuscript, which was developed in the Machine Learning department at Microsoft Research. Not only is this work peer-reviewed but also the implementation in XGBoost is open source software that is available for inspection and independent verification by anyone. For more information please see the documentation: https://xgboost.readthedocs.io/en/latest/tutorials/dart.html.

*9. Lines 159-161: Bayesian optimization for hyperparameter tuning of XGBoost models was performed using the autoxgboost R package (Thomas et al, 2018). The reference points to arXiv. . .. This is a preprint archive (no peer review, nothing). So, what is this? Can we trust?*

> Response: While Bayesian hyperparameter tuning has some advantages, we have updated our code and manuscript in more recent revisions and no longer use the autoxgboost package. Instead, as we explain in our revised methods section, we use 50 sets of potential hyperparameter values that are evaluated for performance in a nested cross-validation (within the training data). An advantage of this approach is that it is considerably faster than the Bayesian optimization and also makes it easier to evaluate the performance of varying combinations of hyperparameters. In our revised manuscript, we have included the following explanation of our updated approach to hyperparameter tuning: "XGBoost has several hyperparameters related to the desired size and complexity of the model that need to be set in training for each dataset. We had a priori selected to tune our XGBoost models with DART using

six hyperparameters (Supplementary Table S2), while using default values for other potential hyperparameters based on previous modelling experience. Our tuning and evaluation approach used two-level (nested) cross-validation. Within each training fold for our outer cross-validation, we further randomly split the training data in half and performed a 2-fold cross-validation to compare the performance of XGBoost models using 50 random sets of potential hyperparameters selected with Latin hypercube sampling (Stein, 1987) to be well-spaced across the range of potential hyperparameter values. While this is more similar to a random search than a grid search, it is expected to more efficiently find well performing sets of hyperparameters than random search, because it decreases the likelihood of checking combinations that are trivially different or leaving unexplored regions in the six-dimensional space, which has too many combinations to effectively cover with a grid search. We selected the set of hyperparameters that minimized the RMSE within the withheld portion of the training data before refitting with all training data."

*10. Lines 175-176. . . The contribution of each feature to cross-validated predictions was estimated by SHAP values (reference. . . arXiv). . . .again this preprint archive. . .*

Response: Again, please see our previous responses on publication practices in the academic field of machine learning where preprints and conference proceedings are reviewed and widely accepted. For example, we cite the preprint as this is the most cited reference for this work but open reviews are also posted at: https://openreview.net/search?term=Consistent+feature+attribution+for+tree+ensembles&content=all&group=ICML.cc/2017/WHI&source=all

*11. Lines 177-227: A lot of information and description is given by the authors, written in a smart appearing way, but it does not help. The reader is lost! He/she just has to believe that everything is ok with this way. But he/she does not trust.*

Response: We have made substantial edits within our introduction, methods, and discussion to add clarity and detail in our description of the machine-learning methods that we employ - particularly as their recent emergence in the field of machine learning means they have not had time yet to be widely adopted into atmospheric sciences. We particularly addressed the specific clarifications sought by referee #1 in their detailed comments (see response and revised manuscript). Again, as in our prior responses, we stress how we present rigorous scientific analyses including multiple approaches to cross-validation and comparison with an independent dataset (SuomiNet CWV). We again emphasize that all of our code and data are archived and are fully reproducible (see our Open Access Zenodo repository DOI: 10.5281/zenodo.3266058) enabling anyone to regenerate our results.

*12. Section 4: results: I avoid to give my comments to the text. . . nobody can check what they state. . ., what is ok, what is not ok, what is trustworthy, what is not trustworthy. There is nothing to judge!*

Response: Please see response #1 and response #11.

*13. Figure 1: There is no hint where we are? no city name, e.g., Boston, New York, no name of any state. . . Maryland. . .. Figure 1 is a nice 'indicator' , . . .. of the feeling I have with the entire paper.*

Response: While we note that our figure includes both latitude and longitude grid lines and labels as well as a descriptive figure legend that explains that the region shown is the Northeastern and Mid-Atlantic USA, we have now added labels for the major urban centers of Boston, New York, and Washington D.C. in our revised manuscript.

*14. Figure 3 and the following figures tell me: MAIAC does a good job, seasonally dependent uncertainties are visible. This is ok, surface properties change and are not perfectly considered in the retrieval. One should accept that. Machine learning procedures may purge the deviations in this specific 'learning region' of Northeast USA. But for any new region . . ..? We have to start again, I believe.*

Response: As we make clear in our discussion of limitations, the generalizability of our findings to new regions was outside the scope of the detailed analyses that we present in this manuscript. Our demonstration of improved agreement of MAIAC CWV with an independent dataset of CWV from SuomiNet GWP stations at new locations in the Northeast USA is evidence that the MAIAC CWV retrieval error can be decreased through our empirical approach. As we discuss, our results are not perfect either although the similarity in the resulting RMSE for Aqua and Terra after applying our correction (both improved versus the use of raw MAIAC CWV values) also suggests that we have reached a plateau of what is possible within our approach. We demonstrate our results over a large region of the United States over 16 years including all seasons, and we make our code and data available for anyone who wants to apply our methods to new regions.

**Gradient Boosting Machine Learning to Improve Satellite-Derived Column Water Vapor Measurement Error**

Allan C. Just[1*], Yang Liu[1], Meytar Sorek-Hamer[2,3], Johnathan Rush[1], Michael Dorman[4], Robert Chatfield[3], Yujie Wang[5,6], Alexei Lyapustin[6], Itai Kloog[4]

5   [1]Icahn School of Medicine at Mount Sinai, New York, USA
    [2]Universities Space Research Association (USRA), California, USA
    [3]NASA Ames Research Center, California, USA
    [4]Ben-Gurion University of the Negev, Beer Sheva, Israel
    [5]University of Maryland Baltimore County, Maryland, USA
10  [6]NASA Goddard Space Flight Center, Maryland, USA

*Correspondence to: Allan C. Just (allan.just@mssm.edu)*

**Abstract.** The atmospheric products of the Multi-Angle Implementation of Atmospheric Correction (MAIAC) algorithm include column water vapor (CWV) at 1 km resolution, derived from daily overpasses of NASA's Moderate Resolution Imaging Spectroradiometer (MODIS)
15  instruments aboard the Aqua and Terra satellites. We have recently shown that machine learning using extreme gradient boosting (XGBoost) can improve the estimation of MAIAC aerosol optical depth (AOD). Although MAIAC CWV is generally well validated (Pearson's R >0.97 versus CWV from AERONET sun photometers), it has not yet been assessed whether machine-learning approaches can further improve CWV. Using a novel spatiotemporal cross-
20  validation approach to avoid overfitting, our XGBoost model with nine features derived from land use terms, date, and ancillary variables from the MAIAC retrieval, quantifies and can correct a substantial portion of measurement error relative to collocated measures at AERONET sites (26.9% and 16.5% decrease in Root Mean Square Error (RMSE) for Terra and Aqua datasets, respectively) in the Northeastern USA, 2000-2015. We use machine-learning
25  interpretation tools to illustrate complex patterns of measurement error and describe a positive bias in MAIAC Terra CWV worsening in recent summertime conditions. We validate our predictive model on MAIAC CWV estimates at independent stations from the SuomiNet GPS network where our corrections decrease the RMSE by 19.7% and 9.5% for Terra and Aqua MAIAC CWV. Empirically correcting for measurement error with machine-learning algorithms is
30  a post-processing opportunity to improve satellite-derived CWV data for Earth science and remote sensing applications.

**1 Introduction**

Water vapor represents a small but environmentally significant constituent of the atmosphere. The integrated water vapor from ground to space is defined as the column water vapor (CWV), in units of centimeters (i.e. precipitable water vapor) (Gao and Goetz, 1990). CWV has important applications in many fields, such as atmospheric correction of remote sensing images, Earth energy balance and global climate change, land surface temperature retrieval in thermal remote sensing, and astronomy. Thus, high resolution CWV values with global coverage have multiple uses in Earth science and remote sensing. CWV has been measured by multiple technologies and monitoring networks, including sun photometers, GPS sensors (e.g. SuomiNet), Aerosol Robotic Network (AERONET) sun photometers, and satellite remote-sensing. The AERONET sun photometer network measures CWV in approximately 400 stations worldwide, in channels centered at 940nm and provided to the user in Level 2, which is the highest data-quality level provided by AERONET (Pérez-Ramírez et al., 2014). The AERONET CWV data have been well validated with the U.S. Department of Energy Atmospheric Radiation Measurement Program (ARM) radiosonde observations and other ground-based retrieval techniques such as microwave radiometry (MWR) and SuomiNet GPS receivers, and do not observe any dependence of biases with the zenith angle (Pérez-Ramírez et al., 2014). AERONET CWV has been used in studies that examine aerosol optical, microphysical, and radiative properties in Africa (Adesina et al., 2014; Boiyo et al., 2019; Kumar et al., 2013), in the Brazilian tropics (Schafer et al., 2008) , and in Beijing and Kanpur (Wang et al., 2011). Global satellite-borne CWV is available at high resolution (1 km), from the Multi-Angle Implementation of Atmospheric Correction (MAIAC) algorithm derived from daily overpasses of NASA's Moderate Resolution Imaging Spectroradiometer (MODIS) instruments aboard the Aqua and Terra satellites. The MAIAC CWV is computed using MODIS near-infrared (NIR) channels centered at 940nm. This method applies two ratios of channels to compute the water vapor transmittance, and then compute the amount of water vapor using look-up-tables (Lyapustin et al., 2014). The MAIAC CWV algorithm was validated against ground measurements of CWV from 265 AERONET stations worldwide, with a relatively strong association (Pearson's R > 0.95; root mean squared error [RMSE] < 0.25 cm; average accuracy of ±15%) (Martins et al., 2019). These datasets were collocated by averaging MAIAC values within 9 x 9 pixels and AERONET values ±30 minutes of the satellite overpass in cloud-free conditions. A significant upward trend (p < .05) for MAIAC TERRA was found over most regions, although this was not significant over the Northeastern USA. Globally, the highest average correlation between MAIAC CWV retrievals from both Aqua and Terra with AERONET CWV have been shown in Asia and both Northern and Southern regions of the USA.

In spite of the strong performance of MAIAC CWV in multiple locations, comparing it with collocated AERONET CWV, there may be opportunities to characterize and correct complex interactions and challenging conditions that increase satellite retrieval error. However, it has not yet been assessed whether machine-learning approaches can improve the estimation of satellite-borne CWV. We have recently demonstrated that machine learning using eXtreme Gradient Boosting (XGBoost) (Chen and Guestrin, 2016) can improve the estimation of MAIAC aerosol optical depth (AOD) parameters over AERONET stations (43% decrease in cross-validated RMSE) (Just et al., 2018). For an introduction to gradient boosted regression trees, please see the work of Elith et al. (2008). XGBoost involves fitting a large number of tree-based models. Each subsequent tree is fit to the error from the previous trees and the predictions of all the trees are added together. Each tree's prediction is multiplied by a shrinkage factor (or "learning rate") η, a number between 0 and 1. By adding successive trees, XGBoost descends the gradient of the loss function. The component trees use a recursive binary partitioning of the predictors that accommodates varying types and scales of predictor variables and is robust to outliers (Elith et al., 2008). An advantage of flexible algorithmic machine-learning approaches such as XGBoost is that they can model complex phenomena (Chen and He, 2015), including interactions of multiple features (e.g. retrieval angles, seasonality, and surface characteristics). The resulting prediction model can be used as an algorithm to reduce the retrieval error. Machine-learning tools for model interpretation can also help explain the contributions of these features to retrieval error and guide feature selection to build parsimonious models.

While the satellite data record continues to grow, the ground monitoring networks that can be used for validation and algorithmic measurement error correction of satellite retrieval products are still sparse. Collocated ground-satellite datasets may thus have important non-independent spatiotemporal structure if they rely on observations that occur in only a few locations. Flexible machine-learning models would overfit to the characteristics of these particular stations or the days when AERONET data are available if cross-validation assumed independence of observations. While machine-learning applications in aerosol research have begun to adopt group-K-fold cross validation for assessing model fit across fixed monitoring networks (Di et al., 2016), we propose a novel cross-validation approach taking into consideration data structure due to both fixed sites and correlation of observations from the same day.

The goals of this work are to 1) evaluate whether machine-learning gradient boosting models can improve satellite-based CWV retrievals, and 2) understand the contributions of different features as well as spatial and temporal structures of the ground station measures to

Author

Author

Author

Author

Author

Author

Author

Author

Author

Author

Author

predictions of error in the estimated CWV. The data and machine-learning methods are described in section 2, followed by a discussion of the results in sections 3 and 4.

**2 Description of the Data**

120 In order to assess the agreement of the MAIAC estimates of CWV with those from AERONET, collocated datasets were built using MAIAC data where AOD was available (representing clear sky conditions) from both Terra and Aqua (separately) collocated to the nearest 1 km * 1 km grid centroid and the closest observation in time (no more than 60 minutes) with cloud-screened (version 2, level 2.0) (Smirnov et al., 2000) measures of CWV from the AERONET

125 network of sun photometers over the Northeastern USA (including 13 states and the District of Columbia from Maine to Virginia). The study period included 10,247 observations (from 75 AERONET stations) for Terra (2000-2015) and 8,536 observations (from 71 stations) for Aqua (2002-2015). All analyses were performed for Terra and Aqua datasets separately. AERONET stations in the Northeast are largely urban and coastal (Fig. 1). We defined our target modeling

130 parameter as the difference between MAIAC and AERONET CWV ($\Delta$CWV = MAIAC CWV - AERONET CWV) such that any variation from zero indicated a component of measurement error that we sought to explain.

After exploratory scatterplots of $\Delta$CWV versus time showed a temporal cluster of large outliers

135 coming from a single AERONET station (City College of New York), observations from this site between 2007-06-17 and 2009-01-01 were dropped from further analysis, including 99 observations collocated for Terra and 95 observations for Aqua datasets. This particular period, which was flanked on both sides with months without values at that station, showed a clear deviation from the monitor's typical trend across the remainder of the study period.

140

The date range for the collocated Terra dataset was from 2000-02-25 to 2015-12-27, including observations from 3,024 unique days (52% of days during this interval). The date range for the collocated Aqua dataset was from 2002-07-04 to 2015-12-28, including observations from 2,627 unique days (53% of days during this interval).

**3 Statistical Methods**

We examined the use of XGBoost (Chen and Guestrin, 2016) for improving satellite-based MAIAC CWV retrievals and decreasing estimation error, as this method had previously outperformed two related supervised learning approaches using regression trees, namely gradient boosting and random forests, in a similar application (Just et al., 2018). The XGBoost algorithm is a popular implementation of boosted regression trees (Friedman, 2001). For an introduction to regression trees, see Strobl et al. (Strobl et al., 2009). A regression tree is a model that specifies recursive binary splits of predictors and assigns a constant value to all cases that end up in the same terminal node (namely, their mean on the dependent variable). The algorithm chooses the splits across all predictors that minimize the variance of the residuals. The maximum number of splits within each tree (also known as the maximum depth) can be set as a hyperparameter. A set of multiple trees can be used for prediction by combining the outputs of the individual trees for each case. Such a set of trees can accommodate complex relationships including non-linearities and interactions while being robust to outliers. Boosting is a method of fitting a series of models iteratively, with each model fit on the residuals of the previous models. While each tree may individually perform relatively poorly at predicting the outcome (and thus is known as a "weak learner"), the combination of many trees can collectively describe complex relationships and account for the impact of many predictors. Further, because boosting includes sequentially learning by combining many iteratively fit trees that address the error in previous trees, this technique performs well, achieving low testing error. The XGBoost package is a scalable gradient boosting implementation with additional features including penalties to avoid overfitting and optimized computational speed (Chen and Guestrin, 2016).

We end up with more parsimonious XGBoost models, i.e. fewer trees, by adopting the concept of 'dropout' from deep learning, in which individual learners are randomly dropped during training. Specifically, we used Dropout meets Additive Regression Trees (DART) (Rashmi and Gilad-Bachrach, 2015). Dropping trees helps to avoid the diminishing contributions and over-specialization of later trees in XGBoost. This is particularly important in our application given the low number of AERONET stations and relatively small size of the collocated datasets for machine-learning algorithms. XGBoost has several hyperparameters related to the desired size and complexity of the model that need to be set in training for each dataset. We had *a priori* selected to tune our XGBoost models with DART using six hyperparameters (Supplementary Table S2), while using default values for other potential hyperparameters based on previous modelling experience. Our tuning and evaluation approach used two-level (nested) cross-

Author — Deleted: ,
Author — Deleted: ,
Author — Deleted: a few
Author — Deleted: for
Author — Deleted: and may vary between applications
Author — Deleted: simple
Author — Deleted: is
Author — Deleted: a weak learner
Author — Deleted: is process of
Author — Deleted: XGBoost models were fit with tree dropout using
Author — Deleted: : Dropout meets Additive Regression Trees
Author — Deleted: The tree dropout implementation in DART
Author — Deleted: s
Author — Deleted: gradient boosted regression trees
Author — Deleted: ,
Author — Deleted: ,
Allan Just 3/10/2020 10:45 AM — Deleted: eight

validation. Within each training fold for our outer cross-validation, we further randomly split the training data in half and performed a 2-fold cross-validation to compare the performance of XGBoost models using 50 random sets of potential hyperparameters selected with Latin hypercube sampling (Stein, 1987) to be well-spaced across the range of potential hyperparameter values. While this is more similar to a random search than a grid search, it is expected to more efficiently find well performing sets of hyperparameters than random search, because it decreases the likelihood of checking combinations that are trivially different or leaving unexplored regions in the six-dimensional space, which has too many combinations to effectively cover with a grid search. 
[revised manuscript text omitted]

540  Chen, T. and He, T.: Higgs boson discovery with boosted trees., NIPS 2014 workshop on high-energy physics and machine learning, 69–80, 2015.

Di, Q., Kloog, I., Koutrakis, P., Lyapustin, A., Wang, Y. and Schwartz, J.: Assessing PM2.5 Exposures with High Spatiotemporal Resolution across the Continental United States., Environ. Sci. Technol., 50(9), 4712–4721, doi:10.1021/acs.est.5b06121, 2016.

545  Elith, J., Leathwick, J. R. and Hastie, T.: A working guide to boosted regression trees., J. Anim. Ecol., 77(4), 802–813, doi:10.1111/j.1365-2656.2008.01390.x, 2008.

Friedman, J. H.: Greedy function approximation: a gradient boosting machine., Ann. Statist., 29(5), 1189–1232, doi:10.1214/aos/1013203451, 2001.

Gao, B.-C. and Goetz, A. F. H.: Column atmospheric water vapor and vegetation liquid water
550  retrievals from Airborne Imaging Spectrometer data, J. Geophys. Res., 95(D4), 3549, doi:10.1029/JD095iD04p03549, 1990.

Just, A. C., De Carli, M. M., Shtein, A., Dorman, M., Lyapustin, A. and Kloog, I.: Correcting Measurement Error in Satellite Aerosol Optical Depth with Machine Learning for Modeling PM2.5 in the Northeastern USA., Remote Sens (Basel), 10(5), doi:10.3390/rs10050803, 2018.

555  Kumar, K. R., Sivakumar, V., Reddy, R. R., Gopal, K. R. and Adesina, A. J.: Inferring wavelength dependence of AOD and Ångström exponent over a sub-tropical station in South Africa using AERONET data: influence of meteorology, long-range transport and curvature effect., Sci. Total Environ., 461-462, 397–408, doi:10.1016/j.scitotenv.2013.04.095, 2013.

Lundberg, S. M., Erion, G. G. and Lee, S.-I.: Consistent Individualized Feature Attribution for
560  Tree Ensembles, arXiv, arXiv:1802.03888, 2018.

Lyapustin, A., Alexander, M. J., Ott, L., Molod, A., Holben, B., Susskind, J. and Wang, Y.: Observation of mountain lee waves with MODIS NIR column water vapor, Geophys. Res. Lett., 41(2), 710–716, doi:10.1002/2013GL058770, 2014.

Martins, V. S., Lyapustin, A., de Carvalho, L. A. S., Barbosa, C. C. F. and Novo, E. M. L. M.:
565  Validation of high-resolution MAIAC aerosol product over South America, J. Geophys. Res. Atmos., 122(14), 7537–7559, doi:10.1002/2016JD026301, 2017.

Martins, V. S., Lyapustin, A., Wang, Y., Giles, D. M., Smirnov, A., Slutsker, I. and Korkin, S.: Global validation of columnar water vapor derived from EOS MODIS-MAIAC algorithm against the ground-based AERONET observations, Atmos. Res., 225, 181–192,
570  doi:10.1016/j.atmosres.2019.04.005, 2019.

Pérez-Ramírez, D., Whiteman, D. N., Smirnov, A., Lyamani, H., Holben, B. N., Pinker, R., Andrade, M. and Alados-Arboledas, L.: Evaluation of AERONET precipitable water vapor versus microwave radiometry, GPS, and radiosondes at ARM sites, J. Geophys. Res. Atmos., 119(15), 9596–9613, doi:10.1002/2014JD021730, 2014.

575 Rashmi, K. V. and Gilad-Bachrach, R.: DART: Dropouts meet Multiple Additive Regression Trees., PMLR, 38, 489–497, 2015.

Schafer, J. S., Eck, T. F., Holben, B. N., Artaxo, P. and Duarte, A. F.: Characterization of the optical properties of atmospheric aerosols in Amazônia from long-term AERONET monitoring (1993–1995 and 1999–2006), J. Geophys. Res., 113(D4), doi:10.1029/2007JD009319, 2008.

580 Smirnov, A., Holben, B. N., Eck, T. F., Dubovik, O. and Slutsker, I.: Cloud-Screening and Quality Control Algorithms for the AERONET Database, Remote Sens. Environ., 73(3), 337–349, doi:10.1016/S0034-4257(00)00109-7, 2000.

Stein, M.: Large Sample Properties of Simulations Using Latin Hypercube Sampling, Technometrics, 29(2), 143–151, doi:10.1080/00401706.1987.10488205, 1987.

585 Strobl, C., Malley, J. and Tutz, G.: An introduction to recursive partitioning: rationale, application, and characteristics of classification and regression trees, bagging, and random forests., Psychol. Methods, 14(4), 323–348, doi:10.1037/a0016973, 2009.

[revised manuscript text omitted]